# Error Sensitivity Modulation based Experience Replay: Mitigating Abrupt Representation Drift in Continual Learning

**Fahad Sarfraz**[*,1], **Elahe Arani**[‡,2] **& Bahram Zonooz**[1,2]
[1]Advanced Research Lab, NavInfo Europe, Netherlands
[2]Dep. of Mathematics and Computer Science, Eindhoven University of Technology, Netherlands
fahad.sarfraz@navinfo.eu, {e.arani, bahram.zonooz}@gmail.com

## Abstract

Humans excel at lifelong learning, as the brain has evolved to be robust to distribution shifts and noise in our ever-changing environment. Deep neural networks (DNNs), however, exhibit catastrophic forgetting and the learned representations drift drastically as they encounter a new task. This alludes to a different error-based learning mechanism in the brain. Unlike DNNs, where learning scales linearly with the magnitude of the error, the sensitivity to errors in the brain decreases as a function of their magnitude. To this end, we propose *ESMER* which employs a principled mechanism to modulate error sensitivity in a dual-memory rehearsal-based system. Concretely, it maintains a memory of past errors and uses it to modify the learning dynamics so that the model learns more from small consistent errors compared to large sudden errors. We also propose *Error-Sensitive Reservoir Sampling* to maintain episodic memory, which leverages the error history to pre-select low-loss samples as candidates for the buffer, which are better suited for retaining information. Empirical results show that ESMER effectively reduces forgetting and abrupt drift in representations at the task boundary by gradually adapting to the new task while consolidating knowledge. Remarkably, it also enables the model to learn under high levels of label noise, which is ubiquitous in real-world data streams. *Code: https://github.com/NeurAI-Lab/ESMER*

## 1 Introduction

The human brain has evolved to engage with and learn from an ever-changing and noisy environment, enabling humans to excel at lifelong learning. This requires it to be robust to varying degrees of distribution shifts and noise to acquire, consolidate, and transfer knowledge under uncertainty. DNNs, on the other hand, are inherently designed for batch learning from a static data distribution and therefore exhibit catastrophic forgetting (McCloskey & Cohen, 1989) of previous tasks when learning tasks sequentially from a continuous stream of data. The significant gap between the lifelong learning capabilities of humans and DNNs suggests that the brain relies on fundamentally different error-based learning mechanisms.

Among the different approaches to enabling continual learning (CL) in DNNs (Parisi et al., 2019), methods inspired by replay of past activations in the brain have shown promise in reducing forgetting in challenging and more realistic scenarios (Hayes et al., 2021; Farquhar & Gal, 2018; van de Ven & Tolias, 2019). They, however, struggle to approximate the joint distribution of tasks with a small buffer, and the model may undergo a drastic drift in representations when there is a considerable distribution shift, leading to forgetting. In particular, when a new set of classes is introduced, the new samples are poorly dispersed in the representation space, and initial model updates significantly perturb the representations of previously learned classes (Caccia et al., 2021). This is even more pronounced in the lower buffer regime, where it is increasingly challenging for the model to recover from the initial disruption. Therefore, it is critical for a CL agent to mitigate the abrupt drift in representations and gradually adapt to the new task.

---

[*]Contributed equally.

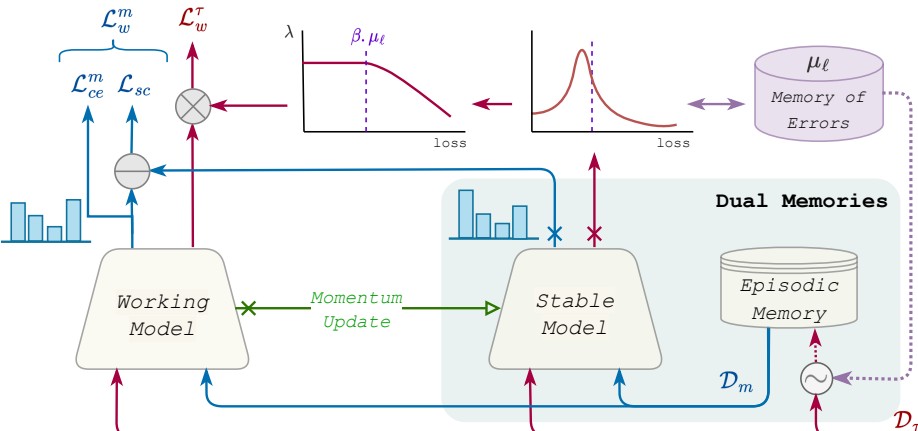

Figure 1: ESMER employs a principled mechanism for modulating the error sensitivity in a dual-memory rehearsal-based system. It includes a stable model which accumulates the structural knowledge in the working model and an episodic memory. Additionally, a memory of errors is maintained which informs the contribution of each sample in the incoming batch towards learning such that the working model learns more from low errors. The stable model is utilized to retain the relational structure of the learned classes. Finally, we employ error sensitive reservoir sampling which uses the error memory to prioritize the representation of low-loss samples in the buffer.

To this end, we look deeper into the dynamics of error-based learning in the brain. Evidence suggests that different characteristics of the error, including its size, affect how the learning process occurs in the brain (Criscimagna-Hemminger et al., 2010). In particular, sensitivity to errors decreases as a function of their magnitude, causing the brain to learn more from small errors compared to large errors (Marko et al., 2012; Castro et al., 2014). This sensitivity is modulated through a principled mechanism that takes into account the history of past errors (Herzfeld et al., 2014) which suggests that the brain maintains an additional memory of errors. The robustness of the brain to high degrees of distribution shifts and its proficiency in learning under uncertainty and noise may be attributed to the principled modulation of error sensitivity and the consequent learning from low errors.

DNNs, on the other hand, lack any mechanism to modulate the error sensitivity and learning scales linearly with the error size. This is particularly troublesome for CL, where abrupt distribution shifts initially cause a considerable spike in errors. These significantly larger errors associated with the samples from unobserved classes dominate the gradient updates and cause disruption of previously learned representations, especially in the absence of sufficient memory samples. To this end, we propose an *Error Sensitivity Modulation based Experience Replay (ESMER)* method that employs a principled mechanism to modulate sensitivity to errors based on its consistency with memory of past errors (Figure 1). Concretely, our method maintains a memory of errors along the training trajectory and utilizes it to adjust the contribution of each incoming sample to learning based on how far they are from the mean statistics of error memory. This allows the model to learn more from small consistent errors compared to large sudden errors, thus gradually adapting to the new task and mitigating the abrupt drift in representations at the task boundary. To keep the error memory stable, task boundary information is utilized to prevent sudden changes. Additionally, we propose an *Error-Sensitive Reservoir Sampling* approach for maintaining the buffer, which utilizes the error memory to pre-select low-loss samples from the current batch as candidates for being represented in the buffer. It ensures that only the incoming samples that have been well learned are added to the buffer that are better suited to retain information and do not cause representation drift when replaying them. The proposed sampling approach also ensures higher-quality representative samples in memory by filtering out outliers and noisy labels, which can degrade performance.

Another salient component of the learning machinery of the brain is the efficient use of multiple memory systems that operate on different timescales (Hassabis et al., 2017; Kumaran et al., 2016). Furthermore, replay of previous neural activation patterns is considered to facilitate memory formation and consolidation (Walker & Stickgold, 2004). These components may play a role in facilitating

lifelong learning in the brain and addressing the challenges of distribution shifts. Therefore, ESMER also employs experience replay in a dual-memory system. In addition to episodic memory, we maintain a semantic memory, which gradually aggregates the knowledge encoded in the weights of the working model and builds consolidated representations, which can generalize across the tasks. The two memories interact to enforce consistency in the functional space and encourage the model to adapt its representations and decision boundary to the new classes while preserving the relational structure of previous classes. The gradual aggregation of knowledge in semantic memory and the enforced relational consistency further aid in mitigating the abrupt change in representations.

We extensively evaluate our method against state-of-the-art rehearsal-based approaches under various CL settings, which simulate different complexities in the real world including distribution shift, recurring classes, data imbalance, and varying degrees of noisy labels. ESMER provides considerable performance improvement in all scenarios, especially with a lower memory budget, demonstrating the versatility of our approach. On the challenging Seq-CIFAR10 with 200 buffer size and 50% label noise, ESMER provides more than 116% gain compared to the baseline ER method. Furthermore, we show that it mitigates abrupt representation drift, is more robust to label noise, and reduces the task recency bias, leading to uniform and stable performance across tasks.

## 2 RELATED WORK

Different approaches have been proposed to address the problem of catastrophic forgetting in DNNs (De Lange et al., 2021). Among them, rehearsal-based methods (Hayes et al., 2021) that use episodic memory for continual rehearsal of samples from previous tasks have shown promise in challenging CL scenarios (Farquhar & Gal, 2018). The base Experience Replay (ER; Riemer et al. (2018)) performs interleaved training of the new task and the memory sample to approximate the joint distribution of tasks. Dark Experience Replay (DER++; Buzzega et al. (2020)) applies an additional distillation loss by storing the output logits together with the samples and enforcing consistency in the output space. Although promising, these methods struggle to approximate the joint distribution of tasks. Caccia et al. (2021) show that the task transition causes a drastic drift in the learned representations of the previous classes seen and propose an asymmetric cross-entropy loss (ER-ACE) on the incoming samples that only considers logits of the new classes. However, an optimal approach to replaying and constraining the model update to retain knowledge and mitigate representation drift remains an open question.

Another promising direction is the utilization of multiple memory systems (Wang et al., 2022; Pham et al., 2021). Notably, CLS-ER (Arani et al., 2021) mimics the interplay between fast and slow learning systems by maintaining two semantic memories that aggregate the weights of the model at different rates using an exponential moving average. It is still unclear, however, what the optimal approach is to build and utilize multiple memory systems. Our method focuses on experience replay in a dual-memory architecture and demonstrates the benefits of mimicking error sensitivity modulation in the brain. We also advocate for a more comprehensive evaluation that covers the different challenges involved in lifelong learning in the real world. In particular, we also evaluate on noisy CL setting in which the agent has to learn continuously from a noisy data stream.

## 3 METHODOLOGY

We first motivate our study with an overview of the issue of abrupt representation drift in DNNs in the CL setting, followed by inspiration from the learning dynamics in the brain. Finally, we present the details of the different components of our method.

### 3.1 ABRUPT REPRESENTATION DRIFT IN DNNS

The sequential learning of tasks in CL potentially exposes the model to varying degrees of distribution shifts in the input and output space. In particular, the introduction of previously unobserved classes at the task boundary may lead to an abrupt representation drift, leading to forgetting. Caccia et al. (2021) show that the samples of the new classes are poorly dispersed and lie near and within the representation space of the previous classes. Therefore, the initial parameter updates cause a significant perturbation of the representations of the previous classes. This is exacerbated by the

inherent imbalance between the samples of the new classes in the current task and the stored samples of previous classes in memory, especially in the lower buffer size regime. Caccia et al. (2021) further show that while the model is able to recover somewhat from the disruptive parameter updates at the task transition with larger buffer sizes, the model fails to recover when the buffer size is small. Moreover, learning scales linearly with the size of the error in standard training. Hence, the considerably higher cross-entropy loss from the unobserved classes dominates learning and biases the gradients towards the new classes. These insights suggest that there is a need for a different learning paradigm tailored for CL that is more robust to abrupt distribution shifts.

### 3.2 INSPIRATION FROM THE BRAIN

The human brain, on the other hand, has evolved to be robust to distribution shifts and to continually learn and interact with a dynamic environment. This is enabled by a complex set of mechanisms and interactions of multiple memory systems. In particular, we probe the dynamics of error-based learning in the brain to draw some insights. Unlike DNNs where learning scales linearly with error size, evidence suggests that the brain modulates its sensitivity to error as a function of error magnitude, so it learns more from small consistent errors compared to large errors (Marko et al., 2012; Castro et al., 2014; Smith & Shadmehr, 2004). To enable such a mechanism, the brain appears to maintain an additional memory of errors that informs error sensitivity modulation (Herzfeld et al., 2014).

Another salient element of the learning machinery of the brain is the efficient use of multiple memory systems that operate on different timescales (Hassabis et al., 2017; Kumaran et al., 2016). Moreover, replay of previous neural activation patterns is considered to facilitate memory formation and consolidation (Walker & Stickgold, 2004; McClelland et al., 1995). These elements could facilitate in enabling lifelong learning in the brain and addressing the challenges of distribution shifts.

### 3.3 PROPOSED APPROACH

We hypothesize that error sensitivity modulation and the interactions of multiple memory systems that operate on different timescales may be crucial for a continual learning agent. To this end, our method aims to incorporate a principled mechanism to modulate error sensitivity based on the history of errors in a dual memory experience replay mechanism. It involves training a working model $\theta_w$ on a sequence of tasks, while maintaining two additional memories: an instance-based fixed-size episodic memory $\mathcal{M}$, which stores input samples from previous tasks, and a parametric stable model $\theta_s$, which gradually aggregates knowledge and builds structured representations.

The CL setting considered here involves learning a sequence of $\mathcal{T}$ i.i.d. tasks from a non-stationary data stream $\mathcal{D}_\tau \in (\mathcal{D}_1, ..., \mathcal{D}_\mathcal{T})$ where the goal is to approximate the joint distribution of all the tasks and distinguish between the observed classes without access to the task identity at inference. In each training step, the model has access to a random batch of labeled incoming samples drawn from the current task $(x_\tau, y_\tau) \sim \mathcal{D}_\tau$ and memory samples drawn from episodic memory $(x_m, y_m) \sim \mathcal{M}$.

#### 3.3.1 LEARNING FROM LOW-LOSS

To mimic the principled mechanism of error sensitivity modulation in the brain, we maintain a memory of errors along the training trajectory and use it to determine how much to learn from each sample. To this end, we maintain error memory using the momentum update of the mean cross-entropy loss $\mu_\ell$ on the batch of samples from the current task. As the stable model is intended to consolidate knowledge and generalize well across tasks, we use it to assess how well the new samples are placed in the consolidated decision boundary and representation space. This also avoids the confirmation bias that can arise from the use of the same learning model.

For each sample $i$ in the current task batch, the cross-entropy loss $\mathcal{L}_{ce}$ is evaluated using stable model:

$$l_s^i = \mathcal{L}_{ce}(f(x_b^i; \theta_s), y_b^i) \tag{1}$$

Subsequently, the weight assigned to each sample for the supervised loss is determined by the distance between the sample loss and the average running estimate $\mu_\ell$:

$$\lambda^i = \begin{cases} 1 & \text{if } l_s^i \leq \beta \cdot \mu_\ell \\ \mu_\ell/l_s^i & \text{otherwise} \end{cases} \tag{2}$$

where $\beta$ controls the margin for a sample to be considered a low-loss. This weighting strategy essentially reduces weight $\lambda^i$ as sample loss moves away from the mean estimate, and consequently reducing learning from high-loss samples so that the model learns more from samples with low-loss. The modulated error-sensitive loss for the working model is then given by the weighted sum of all sample losses in the current task batch: $\mathcal{L}_w^\tau = \sum_i^{|x_b|} \lambda^i \, l_s^i$.

This simple approach to reducing the contributions of large errors can effectively reduce the abrupt drift in representations at the task boundary and allow the model to gradually adapt to the new task. For instance, when previously unseen classes are observed, the loss for their samples would be much higher than the running estimate; thus, by reducing the weights of these samples, the model can gradually adapt to the new classes without disrupting the learned representations of previously seen classes. This implicitly accounts for the inherent imbalance between new task samples and past tasks sample in memory by giving higher weight to samples in memory, which will most likely have a low-loss as they have been learned.

The memory of errors is preserved using a momentum update of loss on the task batch on stable model. To prevent sudden changes when the task switches, we have a task warm-up period during which the running estimate is not updated. This necessitates the task boundary information during training. To further keep the error memory stable and make it robust to outliers in the batch, we evaluate the batch statistics on sample losses which lie within one std of batch loss mean:

$$l_s^f = \{l_s^i \mid i \in \{1, ..., |x_b|\}, \, l_s^i \leq \mu_s + \sigma_s\} \tag{3}$$

where $\mu_s$ and $\sigma_s$ are the mean and std of task losses in stable model, $l_s$. The mean of the filtered sample losses $\bar{l}_s^f$, is then used to update the memory of errors:

$$\mu_\ell \leftarrow \alpha_\ell \, \mu_\ell + (1 - \alpha_\ell) \, \bar{l}_s^f \tag{4}$$

where $\alpha_\ell$ is the decay parameter that controls how quickly the momentum estimate adapts to current values. This provides us with an efficient approach to preserve a memory of errors that informs the modulation of error sensitivity.

### 3.3.2 DUAL MEMORY SYSTEM

To mimic the interplay of multiple memory systems in the brain and employ different learning timescales, our method maintains a parametric semantic memory and a fixed-size episodic memory.

**Episodic Memory** enables the consolidation of knowledge through interleaved learning of samples from previous tasks. We propose *Error-Sensitive Reservoir Sampling* that utilizes the memory of errors to inform Reservoir Sampling (Vitter, 1985). Importantly, we perform a pre-selection of *candidates* for the memory buffer, whereby only the low-loss samples in the current batch of task samples are passed to the buffer for selection:

$$(x_c, y_c) = \{(x_i, y_i) \mid (x_i, y_i) \in \mathcal{D}_\tau, \, l_s^i \leq \beta \cdot \mu_\ell\} \tag{5}$$

This ensures that only the learned samples from the current batch are added to the buffer, thereby preserving the information and reducing the abrupt drift in the learned representations. It can also facilitate the approximation of the distribution of the data stream by filtering out outliers and noise.

**Semantic Memory** is maintained using a parametric model (stable model) that slowly aggregates and consolidates the knowledge encoded in the weights (Krishnan et al., 2019) of the working model. Following Arani et al. (2021), we initialize the stable model with the weights of the working memory, which thereafter keeps a momentum update of that:

$$\theta_s \leftarrow \alpha \theta_s + (1 - \alpha)\theta_w, \quad \text{if } r > u \sim U(0,1) \tag{6}$$

where $\alpha$ is the decay parameter that controls how quickly the semantic model adapts to the working model and $r$ is the update rate to enable stochastic update throughout the training trajectory.

Stable model interacts with episodic memory to implement a dual memory replay mechanism to facilitate knowledge consolidation. It extracts semantic information from buffer samples and uses the relational knowledge encoded in the output logits to enforce consistency in the functional space. This semantic consistency loss encourages the working model to adapt the representation and decision boundary while preserving the relational structure of previous classes. The loss on the memory samples is thus given by the combination of cross-entropy loss and semantic consistency loss:

$$\mathcal{L}_w^m = \mathcal{L}_{ce}(f(X_m; \theta_w), y_m) + \gamma \mathcal{L}_{sc}(f(x_m; \theta_w), f(x_m; \theta_s)) \tag{7}$$

Table 1: Comparison of CL methods in different settings with varying complexities and memory constraints. We report the mean and one std of 3 runs. For completion, we provide results under Task-IL setting by using the task label to select from the subset of output logits from the single head.

| Buffer | Method | Seq-CIFAR10 | | Seq-CIFAR100 | | GCIL | |
|---|---|---|---|---|---|---|---|
| | | Class-IL | Task-IL | Class-IL | Task-IL | Uniform | Longtail |
| – | JOINT | $92.20_{\pm0.15}$ | $98.31_{\pm0.12}$ | $70.62_{\pm0.64}$ | $86.19_{\pm0.43}$ | $58.59_{\pm1.95}$ | $58.42_{\pm1.32}$ |
| | SGD | $19.62_{\pm0.05}$ | $61.02_{\pm3.33}$ | $17.58_{\pm0.04}$ | $40.46_{\pm0.99}$ | $10.38_{\pm0.26}$ | $9.61_{\pm0.19}$ |
| 100 | ER | $41.10_{\pm1.10}$ | $89.34_{\pm1.14}$ | $19.99_{\pm0.45}$ | $54.88_{\pm1.16}$ | $14.43_{\pm0.36}$ | $13.22_{\pm0.60}$ |
| | DER++ | $55.66_{\pm0.78}$ | $89.27_{\pm2.28}$ | $25.25_{\pm2.22}$ | $56.21_{\pm0.74}$ | $21.17_{\pm1.65}$ | $20.29_{\pm1.03}$ |
| | ER-ACE | $52.55_{\pm3.06}$ | $88.96_{\pm1.21}$ | $31.28_{\pm0.36}$ | $59.23_{\pm1.39}$ | $23.76_{\pm1.61}$ | $23.05_{\pm0.18}$ |
| | CLS-ER | $62.30_{\pm1.58}$ | $93.10_{\pm0.72}$ | $38.87_{\pm1.59}$ | $70.29_{\pm0.45}$ | $33.42_{\pm0.30}$ | $33.92_{\pm0.79}$ |
| | ESMER | $\mathbf{65.37}_{\pm0.68}$ | $\mathbf{93.32}_{\pm0.15}$ | $\mathbf{42.89}_{\pm0.40}$ | $\mathbf{73.00}_{\pm0.49}$ | $\mathbf{35.77}_{\pm0.32}$ | $\mathbf{34.26}_{\pm0.26}$ |
| 200 | ER | $44.79_{\pm1.86}$ | $91.19_{\pm0.94}$ | $21.91_{\pm0.36}$ | $61.34_{\pm0.11}$ | $16.52_{\pm0.10}$ | $16.20_{\pm0.30}$ |
| | DER++ | $64.88_{\pm1.17}$ | $91.92_{\pm0.60}$ | $30.68_{\pm1.35}$ | $62.71_{\pm0.68}$ | $27.73_{\pm0.93}$ | $26.48_{\pm2.04}$ |
| | ER-ACE | $62.08_{\pm1.44}$ | $92.20_{\pm0.57}$ | $35.17_{\pm1.17}$ | $63.09_{\pm1.23}$ | $27.44_{\pm0.64}$ | $25.29_{\pm1.89}$ |
| | CLS-ER | $66.10_{\pm0.75}$ | $93.90_{\pm0.60}$ | $43.80_{\pm1.89}$ | $73.49_{\pm1.04}$ | $35.88_{\pm0.41}$ | $35.67_{\pm0.72}$ |
| | ESMER | $\mathbf{69.16}_{\pm0.54}$ | $\mathbf{94.10}_{\pm0.33}$ | $\mathbf{48.77}_{\pm0.31}$ | $\mathbf{75.86}_{\pm0.96}$ | $\mathbf{39.00}_{\pm1.14}$ | $\mathbf{37.98}_{\pm0.88}$ |

where the mean squared loss is used as the semantic consistency loss $\mathcal{L}_{sc}$ and $\gamma$ controls the strength of consistency loss. The overall loss of the working model is the sum of the loss on the new task samples and the buffer samples, that is, $\mathcal{L}_w = \mathcal{L}_w^{\tau} + \mathcal{L}_w^{m}$.

Slow aggregation of knowledge in stable model further helps mitigate the abrupt change in representations. Even after potentially disruptive initial updates at the task boundary, stable model retains the representations of the previous classes, and the enforced consistency loss encourages the working model to retain the relational structure among the previous classes and not deviate too much from them. Therefore, the two components complement each other, and the modulation of error sensitivity coupled with the slow acquisition of semantic knowledge reduces the drift in parameters and hence forgetting. For inference, we use stable model, as it builds consolidated representations that generalize well across tasks (Figure S1 and Table S3). See Algorithm 1 for more details.

## 4 CONTINUAL LEARNING SETTINGS

To faithfully gauge progress in the field, it is imperative to evaluate methods on representative experimental settings that measure how well they address the key challenges of CL in the real world. To this end, we evaluate scenarios that adhere to the proposed desideratas (Farquhar & Gal, 2018) and present a different set of challenges. Class Incremental Learning (*Class-IL*) presents the most challenging scenario (van de Ven & Tolias, 2019) where each task presents a new set of disjoint classes and the agent is required to distinguish between all observed classes without access to the task identity. However, the Class-IL setting assumes that the number of classes per task remains constant, classes do not reappear, and the training samples are well-balanced across classes. *Generalized Class-IL (GCIL)* (Mi et al., 2020) alleviates these limitations and offers a more realistic setting where the number of classes, the appearing classes, and their sample sizes for each task are sampled from probabilistic distributions. We further emphasize that noise is ubiquitous in the real world; therefore, the learning agent is required not only to avoid forgetting, but also to learn under noisy labels. We consider the **Noisy-Class-IL** setting, which adds varying degrees of symmetric noise in the Class-IL setting. Details of the datasets used in each setting are provided in Section A.3.

## 5 EMPIRICAL EVALUATION

We compare our method with strong rehearsal-based methods with single memory (ER, DER++, ER-ACE) and multiple memories (CLS-ER) under different CL settings and uniform experimental settings (details in Section A.1 in Appendix). Table 1 provides the results for the Class-IL and GCIL settings with different data complexity and memory constraints.

Table 2: Performance under extremely low buffer sizes on Seq-CIFAR10 and different number of tasks on Seq-CIFAR100 with 200 buffer size. We report the mean and one std of 3 runs.

| Method | Seq-CIFAR10 (Buffer) | | | | Seq-CIFAR100 (#Tasks) | | |
|---|---|---|---|---|---|---|---|
| | 200 | 100 | 50 | 10 | 5 | 10 | 20 |
| ER | $44.79_{\pm1.86}$ | $41.10_{\pm1.10}$ | $32.51_{\pm1.77}$ | $22.20_{\pm1.36}$ | $21.91_{\pm0.36}$ | $14.17_{\pm0.53}$ | $9.97_{\pm0.68}$ |
| DER++ | $64.88_{\pm1.17}$ | $55.66_{\pm0.78}$ | $49.36_{\pm2.12}$ | $29.04_{\pm2.65}$ | $30.68_{\pm1.35}$ | $25.50_{\pm3.07}$ | $20.50_{\pm1.42}$ |
| ER-ACE | $62.08_{\pm1.44}$ | $52.55_{\pm3.06}$ | $43.92_{\pm3.74}$ | $26.26_{\pm2.88}$ | $35.17_{\pm1.17}$ | $25.75_{\pm1.56}$ | $18.68_{\pm0.82}$ |
| CLS-ER | $66.10_{\pm0.75}$ | $62.30_{\pm1.58}$ | $53.80_{\pm3.34}$ | $33.87_{\pm4.71}$ | $43.80_{\pm1.89}$ | $35.42_{\pm0.47}$ | $25.98_{\pm1.70}$ |
| ESMER | $\mathbf{69.16}_{\pm0.54}$ | $\mathbf{65.37}_{\pm0.68}$ | $\mathbf{59.94}_{\pm1.67}$ | $\mathbf{36.83}_{\pm1.53}$ | $\mathbf{48.77}_{\pm0.31}$ | $\mathbf{36.37}_{\pm0.46}$ | $\mathbf{27.26}_{\pm0.52}$ |

Table 3: Effect of varying degrees of label noise on Seq-CIFAR10 and Seq-CIFAR100 datasets. All experiments use a buffer size of 200. We report the mean and one std of 3 runs.

| Method | Seq-CIFAR10 | | | | Seq-CIFAR100 | | | |
|---|---|---|---|---|---|---|---|---|
| | Clean | 10% | 25% | 50% | Clean | 10% | 25% | 50% |
| ER | $44.79_{\pm1.86}$ | $36.97_{\pm1.80}$ | $25.54_{\pm2.14}$ | $19.72_{\pm1.53}$ | $21.91_{\pm0.36}$ | $17.12_{\pm0.36}$ | $13.00_{\pm0.42}$ | $8.01_{\pm0.07}$ |
| DER++ | $64.88_{\pm1.17}$ | $53.10_{\pm3.83}$ | $43.17_{\pm3.40}$ | $32.95_{\pm2.43}$ | $30.68_{\pm1.35}$ | $25.13_{\pm1.07}$ | $18.99_{\pm0.62}$ | $11.04_{\pm0.42}$ |
| ER-ACE | $62.08_{\pm1.44}$ | $50.39_{\pm1.29}$ | $37.16_{\pm1.01}$ | $30.14_{\pm0.44}$ | $35.17_{\pm1.17}$ | $23.77_{\pm1.14}$ | $13.59_{\pm0.66}$ | $5.71_{\pm0.33}$ |
| CLS-ER | $66.10_{\pm0.75}$ | $54.86_{\pm1.86}$ | $34.49_{\pm2.23}$ | $23.97_{\pm1.39}$ | $43.80_{\pm1.89}$ | $35.19_{\pm0.85}$ | $24.59_{\pm0.35}$ | $12.16_{\pm0.69}$ |
| ESMER | $\mathbf{69.16}_{\pm0.54}$ | $\mathbf{63.24}_{\pm1.31}$ | $\mathbf{56.65}_{\pm1.01}$ | $\mathbf{42.72}_{\pm2.70}$ | $\mathbf{48.77}_{\pm0.31}$ | $\mathbf{41.80}_{\pm0.88}$ | $\mathbf{33.49}_{\pm1.29}$ | $\mathbf{20.68}_{\pm1.21}$ |

Class-IL requires the method to transfer and consolidate knowledge while also tackling the abrupt representation drift at the task boundary when new classes are observed. The consistent performance improvement in both considered datasets demonstrates the proficiency of ESMER in consolidating knowledge with a low memory budget. Generally, we observe that multiple-memory systems perform considerably better in these challenging settings. Note that Seq-CIFAR100 is a considerably more complex setting in terms of both the recognition task among semantically similar classes and the potential representation drift in the split setting, which introduces 20 unobserved classes with high semantic similarity to previous classes. The improvement of ESMER over the well-performing CLS-ER in this challenging setting suggests that our error-sensitivity modulation method successfully mitigates representation drift and reduces interference among similar tasks. The performance of our method under extremely small buffer sizes in Table 2, which exacerbates abrupt representation drift, further supports the claim. This gain is promising, given that CLS-ER uses two semantic memories with different update frequencies, while our method uses only one.

Table 1 also provides the results for the GCIL setting, which further requires the model to tackle the challenges of class imbalance, sample efficiency, and learn from multiple occurrences of an abject. ESMER provides a consistent improvement in both uniform and longtail settings. The error-sensitivity modulation and consequent gradual adaptation to unobserved classes can allow the model to deal with the class imbalance and learn over multiple occurrences well. Considering a new task with a mix of unobserved and recurring classes, the loss of samples from the recurring classes would most likely be low, as they have been learned earlier, while the loss on unobserved classes would be significantly higher. Consequently, the contribution of unobserved classes would be down-weighted, allowing the model to focus and adapt the representations of the recurring class with reduced interference and representation drift from the unobserved classes. Learning more from low errors also provides more consistent learning and can improve sample efficiency by avoiding potentially suboptimal gradient directions caused by outliers.

Finally, Table 3 provides the results under the Noisy-Class-IL setting where the agent is required to concurrently tackle the challenges of learning from noisy labels. Our strong empirical results demonstrate the superiority of ESMER in tackling the unique challenges of this setting. Learning from low errors implicitly makes our model more robust to label noise and reduces memorization since noisy labels would correspond to higher errors. Furthermore, the error-sensitive reservoir sampling approach ensures a higher purity of the memory buffer, which plays a critical role in noisy CL as buffer samples are replayed multiple times, further aggravating their harmful effect.

Table 4: **Ablation Study:** Effect of systematically removing different components of ESMER on performance on Seq-CIFAR10 with different buffer sizes and label noise (for 200 buffer size).

| Error Sensitivity Modulation | Stable Model | Error Sensitive Reservoir Sampling | Buffer | | | Label Noise | |
|:---:|:---:|:---:|:---:|:---:|:---:|:---:|:---:|
| | | | 50 | 100 | 200 | 25% | 50% |
| ✓ | ✓ | ✓ | **59.94**±1.67 | **65.37**±0.68 | **69.16**±0.54 | **56.61**±1.40 | **42.43**±2.89 |
| ✓ | ✓ | ✗ | 57.53±1.80 | 64.44±0.62 | 69.07±0.87 | 51.16±1.56 | 34.42±2.79 |
| ✗ | ✓ | ✗ | 37.11±4.88 | 51.59±5.26 | 63.41±1.43 | 38.84±0.52 | 22.10±2.62 |
| ✓ | ✗ | ✗ | 36.03±0.83 | 42.56±0.56 | 54.58±2.89 | 38.50±1.53 | 24.70±1.49 |
| ✗ | ✗ | ✗ | 32.51±1.77 | 41.10±0.10 | 44.79±1.86 | 25.54±2.14 | 19.72±1.53 |

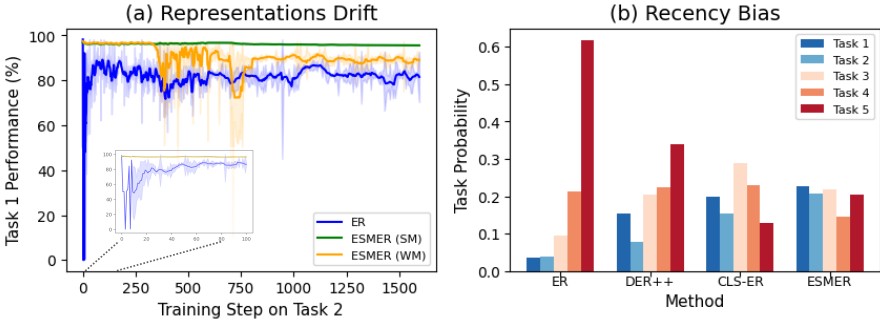

Figure 2: (a) Effect on Task 1 performance as the models are trained on Task 2. SM and WM denote stable model and working model (b) The average probability of predicting each task at the end of training on Seq-CIFAR10 with 200 buffer size. Figure S2 provides recency bias for other settings.

## 6 WHY DOES ESMER WORK?

Here, we attempt to provide insights into what enables the gains with ESMER. We perform our analysis on Seq-CIFAR10 with 200 buffer size and compare it with ER, DER++ and CLSER.

**Ablation Study:** We systematically remove the salient components of our method and evaluate their effect on performance in Table 4. Error-sensitivity modulation provides considerable gains in both low-buffer and noisy label settings. Furthermore, the gradual accumulation of knowledge in stable model proves to be a highly proficient approach to consolidating knowledge. Finally, the proposed error-sensitive reservoir sampling technique provides further generalization gains, particularly in the Noisy-Class-IL setting, where maintaining a clean buffer is critical. Importantly, these components complement each other so that the combination provides the maximum gains.

**Drift in Representations:** To assess ESMER in mitigating the abrupt drift in representations at the task boundary, we monitor the performance of the model on Task 1 as it learns Task 2. Figure 2 (a) shows that ESMER successfully maintains its performance, while ER suffers a sudden performance drop and fails to recover from earlier disruptive updates. Furthermore, while the working model in ESMER undergoes a reduced representation drift after the task warmup period, the stable model retains the performance and provides a stable reference to constrain the update of the working model, helping it recover from the disruption. This suggests that error-sensitivity modulation, coupled with gradual adaptation in stable model, is a promising approach to mitigate abrupt representation drift.

**Robustness to Noisy Labels:** We probe into what enables ESMER to be more robust to label noise. One of the key challenges in learning under noisy labels is the tendency of DNNs to memorize incorrect labels (Arpit et al., 2017) that adversely affect their performance (Sukhbaatar et al., 2014). To evaluate the degree of memorization, we track the performance of the model on noisy and clean samples during training. Figure 3 (a) shows that ESMER significantly reduces memorization, leading to better performance. Furthermore, the quality of the samples in the memory buffer can play a critical role in the performance of the model, as the memory samples are replayed multiple times. Figure 3 (b) compares the percentage of noisy labels in the buffer with our proposed error-sensitive reservoir sampling and the typical reservoir sampling approach. Our approach considerably reduces the probability of adding a noisy sample in episodic memory and maintains a higher quality of the

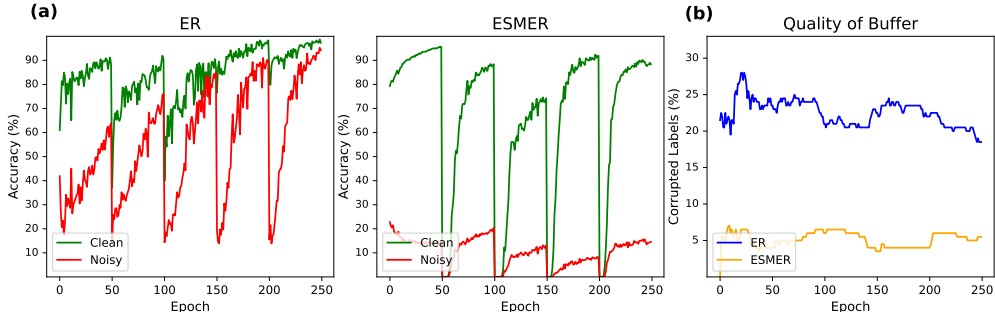

Figure 3: (a) Accuracy of the models on clean and noisy samples as training progresses on Seq-CIFAR10 with 50% label noise. A lower value is better as it indicates less memorization. See Figure S3 for the working model. (b) Percentage of noisy labels in the memory with reservoir sampling in ER and error-sensitive reservoir sampling in ESMER.

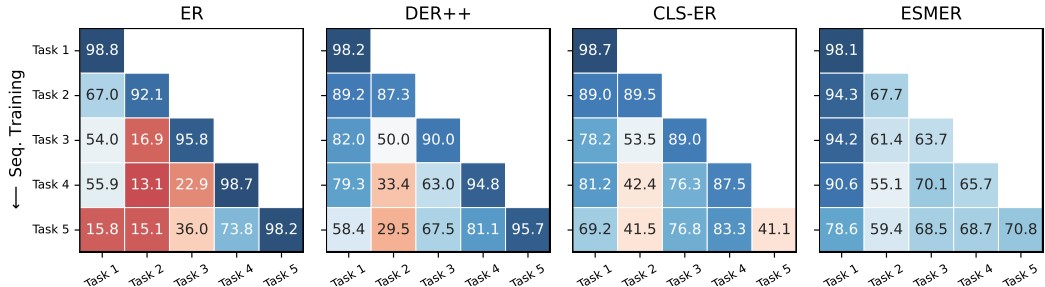

Figure 4: The performance of the models (x-axis) after training on each task in Seq-CIFAR10 with 200 buffer size. See Figure S4 for other datasets and buffer sizes.

buffer. Note that both features are a byproduct of mimicking error-sensitivity modulation in the brain rather than being explicitly designed for the setting.

**Recency Bias:** The sequential learning of tasks with standard SGD in CL induces a recency bias in the model, making it much more likely to predict newly learned classes (Hou et al., 2019). To evaluate the task recency bias of the different methods, we calculate the average probability of predicting each task at the end of training by averaging the softmax outputs of the classes associated with each task on the test set. Figure 2 (b) shows that ESMER provides more uniform predictions across tasks. Generally, methods that employ multiple memories are better at reducing the recency bias, and the dynamics of low-loss learning in our method further reduce the bias. Figure 4 further shows that ESMER maintains a better balance between the plasticity and the stability of the model.

## 7 CONCLUSION

Inspired by the dynamics of error-based learning and multiple memory systems in the brain, we proposed ESMER, a novel approach for error sensitivity modulation based experience replay in a dual memory system. The method maintains a memory of errors along the training trajectory, which is used to modulate the contribution of incoming samples towards learning such that the model learns more from low errors. Additionally, semantic memory is maintained that gradually aggregates the knowledge encoded in the weights of the working model using momentum update. We also proposed an error-sensitive reservoir sampling approach that prioritizes the representation of low-loss samples in the memory buffer, which are better suited for retaining knowledge. Our empirical evaluation demonstrates the effectiveness of our approach in challenging and realistic CL settings. ESMER mitigates abrupt drift in representations, provides higher robustness to noisy labels, and provides more uniform performance across tasks. Our findings suggest that a principled mechanism for learning from low errors in a dual-memory system provides a promising approach to mitigating abrupt representation drift in CL that warrants further study.

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

# A   APPENDIX

---

**Algorithm 1** Error Sensitivity Modulation based Experience Replay (ESMER) Algorithm

---

**Input and Params:** error memory params $\alpha_\ell$ and $\beta$; stable model params $\alpha$ and $r$; semantic consistency weight $\gamma$; data stream $\mathcal{D}$; learning rate $\eta$

**Initialize:** $l_\mu = 0$, $\mathcal{M} \leftarrow \{\}$, $\theta_s = \theta_w$

1: **while** Training **do**
2:     Sample batch from task stream, $(x_\tau, y_\tau) \sim \mathcal{D}_\tau$, and episodic memory, $(x_m, y_m) \sim \mathcal{M}$
3:     Evaluate loss of task samples on stable model, $l_s^i$ (Eq. 1)
4:     Get the weights of task samples, $\lambda^i$ (Eq. 2)
5:     Evaluate the sensitivity-modulated task loss, $\mathcal{L}_w^\tau = \sum_i^{|x_b|} \lambda^i \cdot l_s^i$
6:     Evaluate the cross-entropy and consistency loss on memory samples, $\mathcal{L}_w^m$ (Eq. 7)
7:     Combine overall loss: $\mathcal{L}_w = \mathcal{L}_w^\tau + \mathcal{L}_w^m$
8:     Update parameters of working model: $\theta_w \leftarrow \theta_w - \eta \nabla_{\theta_w} \mathcal{L}_w$
9:     Update the stable model: $\theta_s \leftarrow \alpha \theta_s + (1 - \alpha)\, \theta_w, \quad if \ \ r > a \sim U(0, 1)$   (Eq. 6)
10:     Filter candidates for episodic memory, $(x_c, y_c)$ (Eq. 5)
11:     Update episodic memory: $\mathcal{M} \leftarrow \text{Reservoir}(\mathcal{M}, (x_c, y_c))$
12:     Remove outliers from task batch loss: $l_s^f$ (Eq. 3)
13:     Update error memory if epoch > task warm-up period: $\mu_\ell$ (Eq. 4)
    **return** $\theta_s$

---

## A.1   EXPERIMENTAL DETAILS

We conducted all our experiments under common settings to disentangle the performance improvement of the different CL methods from the training settings (Mirzadeh et al., 2020). Following (Arani et al., 2021; Buzzega et al., 2020), we use ResNet-18 network and train with SGD optimizer. For all our experiments, we train for 50 epochs with a batch size of 32 for both incoming samples and memory and apply random crop and horizontal flip data augmentations. Note that we store non-augmented images in the buffer and apply the data augmentations to the memory samples for replay. We report the mean and std of 3 runs.

### A.1.1   HYPERPARAMETER TUNING

We use a small validation set to select the hyperparameters for all methods. For ESMER, we use $\alpha_l = 0.99$, $\alpha_l = 0.999$, and tune $\gamma \in \{0.15, 2.0\}$, $\beta \in [0.8, 3.0]$ and $r \in [0, 1]$. Note that we did not conduct an exhaustive parameter search for each experiment, and it is possible that a better hyperparameter search can further improve performance. The hyperparameters used for each experiment are provided in Table S1.

For the baselines, we use the results from (Arani et al., 2021) if similar settings are provided; otherwise, we conduct a grid search over the hyperparameter values used in the original paper. For ER and ER-ACE, we search $\eta \in \{0.1, 0.01, 0.001\}$. For DER++ we search over $\eta \in \{0.1, 0.01, 0.001\}$, $\alpha \in \{0.1, 0.2, 0.5, 1.0\}$ and $\beta \in \{0.5, 1.0\}$. Finally, for CLS-ER, we search over $\alpha_s = 0.999$, $\alpha_p = 0.999$, $\lambda \in \{0.15, 2.0\}$, $r_p \in [0, 1]$ and $r_s \in [0, 1]$. The parameters selected for the baselines in each setting are provided in Table S2.

## A.2   COMPARISON WITH NEUROMODULATION

Metaplasticity (Abraham, 2008) and Neuromodulation (Doya, 2002) refers to different mechanisms to modulate the plasticity rate of individual synapses (strengths / weights of the incoming and outgoing connections in DNNs) (Kudithipudi et al., 2022; Bailey et al., 2000). These form the biological underpinning behind the popular regularization-based approaches where the goal is to identify synapses that are important for previous tasks and penalize changes to them while learning a new task (Zenke et al., 2017; Kirkpatrick et al., 2017). Neuromodulation has also been explored in conjunction with local learning (Madireddy et al., 2020) which shows promise.

The error sensitivity modulation in ESMER differs substantially from these methods, as instead of modulating the plasticity rate of individual synapses, we modulate the contribution of each sample

Table S1: Hyperparameters for ESMER. For all experiments, the models are trained for 50 epochs with a batch size of 32 for incoming and memory samples.

| Dataset | Buffer | Label noise | $\eta$ | $\alpha_l$ | $\beta$ | $\gamma$ | $\alpha$ | $r$ |
|---|---|---|---|---|---|---|---|---|
| Seq-CIFAR100 | 100 | 0 | 0.03 | 0.99 | 1.2 | 0.15 | 0.999 | 0.07 |
| | 200 | 0 | 0.03 | 0.99 | 1.0 | 0.15 | 0.999 | 0.07 |
| GCIL - Unif | 100 | 0 | 0.05 | 0.99 | 2.5 | 0.2 | 0.999 | 0.2 |
| | 200 | 0 | 0.05 | 0.99 | 2.5 | 0.2 | 0.999 | 0.2 |
| GCIL - Longtail | 100 | 0 | 0.05 | 0.99 | 3.0 | 0.2 | 0.999 | 0.2 |
| | 200 | 0 | 0.05 | 0.99 | 2.5 | 0.2 | 0.999 | 0.2 |
| Seq-CIFAR10 | 10 | 0 | 0.03 | 0.99 | 0.9 | 0.15 | 0.999 | 0.1 |
| | 50 | 0 | 0.03 | 0.99 | 0.9 | 0.15 | 0.999 | 0.1 |
| | 100 | 0 | 0.03 | 0.99 | 0.9 | 0.15 | 0.999 | 0.1 |
| | 200 | 0 | 0.03 | 0.99 | 1.2 | 0.15 | 0.999 | 0.1 |
| Noisy-Seq-CIFAR10 | 200 | 10% | 0.03 | 0.99 | 0.9 | 0.15 | 0.999 | 0.1 |
| | 200 | 25% | 0.03 | 0.99 | 0.9 | 0.15 | 0.999 | 0.1 |
| | 200 | 50% | 0.03 | 0.99 | 1.0 | 0.15 | 0.999 | 0.1 |

in the batch based on its error magnitude. Our work aims to bring about a salient characteristic of the learning dynamics of the brain: it learns more from small consistent errors compared to large errors. This is not the case for standard ANNs, where learning scales linearly with the error size. Therefore, the error sensitivity modulation in ESMER specifically focuses on utilizing a memory of errors to modulate the contribution of samples to learning based on their error magnitude, so that the model learns more from small errors. Please note that our approach is complementary to metaplasticity and neuromodulation-based approaches, and the interaction of these biologically plausible mechanisms is an interesting avenue of research.

## A.3 DATASETS

Here we provide details of the datasets used in each of the considered CL setting.

### A.3.1 CLASS-IL

For Class-IL, we use the common sequential versions of CIFAR10 (Seq-CIFAR10) and CIFAR100 (Seq-CIFAR100) datasets whereby the classes are split into five tasks each containing a disjoint set of 2 and 20 classes, respectively. We do not vary the order of classes, and so, for instance, Task 1 in Seq-CIFAR10 always contains the first two classes.

Note that our method does not utilize separate classification heads or subnets. For completion, we also evaluate the performance of the same model trained for Class-IL under the Task-IL setting where the model assumes availability of task labels at inference. We use the task label to narrow down predictions to the subset of output logits belonging to classes belonging to the task.

### A.3.2 GCIL

Following (Mi et al., 2020; Arani et al., 2021), we apply the GCIL setting on CIFAR100 dataset. The setting involves training 20 tasks sequentially, and for each task, the number and appearing classes are sampled from a probability distribution but capped at a maximum of 50 and subsequently 1000 samples are drawn from either a uniform (Unif) or longtail sample distributions. For further details, see (Mi et al., 2020).

Importantly, we found that even for a fixed dataset seed, the corresponding tasks in the Longtail and Unif settings differ not only in the sample distribution (as expected) but also significantly in the number of classes drawn and the appearing classes. For instance, with the dataset seed set to 1993, the last task in the Longtail setting contains 50 classes, whereas the corresponding task in the Unif setting contains only 17 classes. This makes the former task significantly easier.

Table S2: Selected hyperparameters for baselines. All methods are trained for 50 epochs with a batch size of 32 for incoming and memory samples. For CLS-ER, $\alpha_s = \alpha_p = 0.999$

| Dataset | Buffer | Method | Hyperparameters |
|---|---|---|---|
| Seq-CIFAR100 | 100 | ER | $\eta$=0.1 |
| | | ER-ACE | $\eta$=0.01 |
| | | DER++ | $\eta$=0.03, $\alpha$=0.1, $\beta$=0.1 |
| | | CLS-ER | $\eta$=0.1 $\lambda$=0.15, $r_p$=0.1, $r_s$=0.05 |
| | 200 | ER | $\eta$=0.1 |
| | | ER-ACE | $\eta$=0.01 |
| | | DER++ | $\eta$=0.03, $\alpha$=0.2, $\beta$=0.1 |
| | | CLS-ER | $\eta$=0.1 $\lambda$=0.15, $r_p$=0.1, $r_s$=0.05 |
| GCIL-Unif | 100 | ER | $\eta$=0.1 |
| | | ER-ACE | $\eta$=0.1 |
| | | DER++ | $\eta$=0.03, $\alpha$=0.5, $\beta$=0.1 |
| | | CLS-ER | $\eta$=0.1 $\lambda$=0.1, $r_p$=0.7, $r_s$=0.09 |
| | 200 | ER | $\eta$=0.1 |
| | | ER-ACE | $\eta$=0.1 |
| | | DER++ | $\eta$=0.03, $\alpha$=0.2, $\beta$=0.1 |
| | | CLS-ER | $\eta$=0.1 $\lambda$=0.1, $r_p$=0.7, $r_s$=0.1 |
| GCIL-Longtail | 100 | ER | $\eta$=0.1 |
| | | ER-ACE | $\eta$=0.1 |
| | | DER++ | $\eta$=0.03, $\alpha$=0.2, $\beta$=0.1 |
| | | CLS-ER | $\eta$=0.1 $\lambda$=0.1, $r_p$=0.7, $r_s$=0.08 |
| | 200 | ER | $\eta$=0.1 |
| | | ER-ACE | $\eta$=0.1 |
| | | DER++ | $\eta$=0.03, $\alpha$=0.5, $\beta$=0.1 |
| | | CLS-ER | $\eta$=0.1 $\lambda$=0.1, $r_p$=0.7, $r_s$=0.1 |
| Seq-CIFAR10 | 10 | ER | $\eta$=0.1 |
| | | ER-ACE | $\eta$=0.01 |
| | | DER++ | $\eta$=0.03, $\alpha$=0.5, $\beta$=0.5 |
| | | CLS-ER | $\eta$=0.1 $\lambda$=0.15, $r_p$=0.3, $r_s$=0.04 |
| | 50 | ER | $\eta$=0.1 |
| | | ER-ACE | $\eta$=0.01 |
| | | DER++ | $\eta$=0.03, $\alpha$=0.2, $\beta$=0.5 |
| | | CLS-ER | $\eta$=0.1 $\lambda$=0.15, $r_p$=0.3, $r_s$=0.05 |
| | 100 | ER | $\eta$=0.1 |
| | | ER-ACE | $\eta$=0.01 |
| | | DER++ | $\eta$=0.03, $\alpha$=0.2, $\beta$=0.5 |
| | | CLS-ER | $\eta$=0.1 $\lambda$=0.15, $r_p$=0.3, $r_s$=0.05 |
| Noisy-Seq-CIFAR10 | 200 | ER | $\eta$=0.1 |
| | | ER-ACE | $\eta$=0.01 |
| | | DER++ | $\eta$=0.03, $\alpha$=0.1, $\beta$=0.5 |
| | | CLS-ER | $\eta$=0.1 $\lambda$=0.15, $r_p$=0.3, $r_s$=0.1 |

We fix this in our implementation by drawing 20 random numbers (after setting the random seed to 1993) when the dataset is initialized and setting the same unique seed before sampling each task. This ensures that the same number of classes and the appearing classes (if not all the samples for a particular class are seen already) are drawn for the two settings. This provides a better evaluation of the effect of the difference in data distributions between the two variants.

This leads to a significant difference in baseline performance compared to those reported in (Arani et al., 2021). We also perform the hyperparameter for all the considered methods and find that, for the modified setting, our choice of parameters performs considerably better than the ones used in (Arani et al., 2021). We report the results on more optimal hyperparameters provided in Table S2.

Table S3: Working model and stable model performance on all the considered CL scenarios. Stable model consistently provides better performance.

| Dataset | Buffer | Label Noise | Stable Model | Working Model |
|---------|--------|-------------|--------------|---------------|
| Seq-CIFAR100 | 100 | - | $42.89_{\pm0.40}$ | $27.56_{\pm1.26}$ |
| | 200 | - | $48.77_{\pm0.31}$ | $37.87_{\pm1.29}$ |
| GCIL - Unif | 100 | - | $35.77_{\pm0.32}$ | $23.36_{\pm0.81}$ |
| | 200 | - | $39.00_{\pm1.14}$ | $29.55_{\pm1.24}$ |
| GCIL - Longtail | 100 | - | $34.26_{\pm0.26}$ | $22.94_{\pm0.28}$ |
| | 200 | - | $37.98_{\pm0.88}$ | $28.59_{\pm0.94}$ |
| Seq-CIFAR10 | 10 | - | $36.83_{\pm1.53}$ | $26.35_{\pm0.89}$ |
| | 50 | - | $59.94_{\pm1.67}$ | $44.98_{\pm3.18}$ |
| | 100 | - | $65.37_{\pm0.68}$ | $55.80_{\pm2.45}$ |
| | 200 | - | $69.16_{\pm0.54}$ | $65.18_{\pm3.14}$ |
| Noisy-Seq-CIFAR10 | 200 | 10% | $63.24_{\pm1.31}$ | $56.01_{\pm1.14}$ |
| | 200 | 25% | $56.65_{\pm1.01}$ | $48.76_{\pm1.57}$ |
| | 200 | 50% | $42.72_{\pm2.70}$ | $34.24_{\pm2.66}$ |

### A.3.3 NOISY-CLASS-IL

Noisy Class-IL aims to evaluate the robustness of a learning agent to label noise. Since noise is ubiquitous in real-world streams, ideally, a CL method should be able to tackle the challenges of learning under noisy labels and CL concurrently. This requires them to avoid memorization and forgetting. We consider varying degrees of symmetric label noise on Seq-CIFAR10 dataset, **Noisy-Seq-CIFAR10**. For each task, we sample random labels from the classes in the task and replace a fraction of the original labels with the random ones. Note that, under this setting, the random label may correspond to the original label.

### A.4 WORKING MODEL PERFORMANCE

To assess how well stable model aggregates knowledge across the task, we compare its performance with the performance of the working model across all datasets and memory constraints in Table S3. Stable model considerably provides higher performance. To better understand, we look at the task-wise performance on the test set at the end of each task in Figure S1. Stable model retains more knowledge of previous tasks and consolidates information to build representations that generalize across tasks.

### A.5 EFFECT OF HYPERPARAMETERS

Tuning ESMER involves setting the hyperparameters for two complementary components: error sensitivity modulation ($\beta$, $\alpha_l$) and stable model ($r$, $\alpha$, $\gamma$). To understand the interaction of these components and the sensitivity of our method to hyperparameters, we look at the performance of the model with different sets of values for the most pertinent parameters ($\beta$, $r$). Table S4 shows that the loss margin, $\beta$, and the update frequency, $r$ complement each other and, generally, a higher $r$ requires a lower $\beta$. Furthermore, a different set of values can provide similar performance, which significantly facilitates hyperparameter tuning. As can be seen in Table S1, the majority of parameters ($\alpha_s$, $\alpha_\ell$ and $\gamma$) can be fixed, and the optimal parameters do not vary much for different memory budgets.

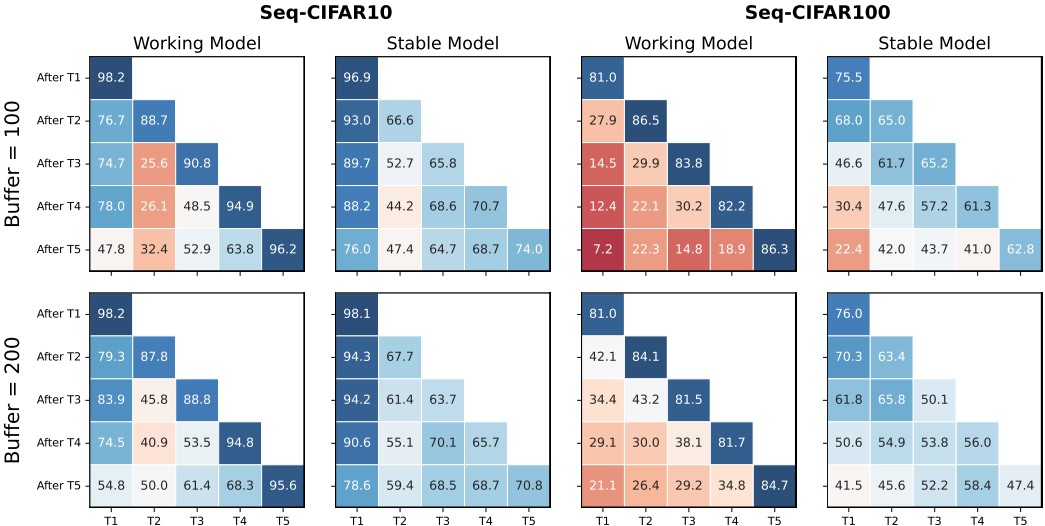

Figure S1: Comparison between the task-wise performance (x-axis) of working model and stable model after training on each task. Stable model provides higher stability and provides more uniform performance across the tasks.

Table S4: Performance of the models trained on Seq-CIFAR10 with different hyperparameters. We use $\eta$=0.03, $\alpha_l$=0.99, $\alpha$=0.999 and $\gamma$=0.15 for all the experiments.

| Buffer | $r$ | $\beta$ | Stable Model | Working Model |
|--------|-----|---------|--------------|---------------|
| 100 | 0.08 | 0.8 | $58.74_{\pm0.77}$ | $53.76_{\pm2.04}$ |
| | | 1.0 | $60.67_{\pm2.40}$ | $54.81_{\pm3.19}$ |
| | | 1.2 | $63.61_{\pm1.13}$ | $54.26_{\pm1.25}$ |
| | 0.10 | 0.8 | $61.21_{\pm1.02}$ | $52.86_{\pm2.03}$ |
| | | 1.0 | $65.37_{\pm0.68}$ | $55.80_{\pm2.45}$ |
| | | 1.2 | $62.73_{\pm2.42}$ | $51.43_{\pm3.90}$ |
| | 0.20 | 0.8 | $59.63_{\pm1.07}$ | $55.96_{\pm1.44}$ |
| | | 1 | $59.29_{\pm1.17}$ | $55.20_{\pm0.97}$ |
| | | 1.2 | $57.49_{\pm2.57}$ | $52.98_{\pm3.45}$ |
| 200 | 0.08 | 0.8 | $59.06_{\pm2.98}$ | $62.15_{\pm1.86}$ |
| | | 1.0 | $62.47_{\pm0.75}$ | $64.34_{\pm0.87}$ |
| | | 1.2 | $63.64_{\pm0.55}$ | $64.00_{\pm0.83}$ |
| | 0.1 | 0.8 | $64.99_{\pm2.57}$ | $63.82_{\pm0.81}$ |
| | | 1 | $66.79_{\pm1.04}$ | $63.78_{\pm1.46}$ |
| | | 1.2 | $69.16_{\pm0.54}$ | $65.18_{\pm3.14}$ |
| | 0.2 | 0.8 | $64.81_{\pm1.50}$ | $61.70_{\pm1.55}$ |
| | | 1 | $67.37_{\pm1.48}$ | $64.40_{\pm1.74}$ |
| | | 1.2 | $66.36_{\pm0.11}$ | $62.93_{\pm0.50}$ |

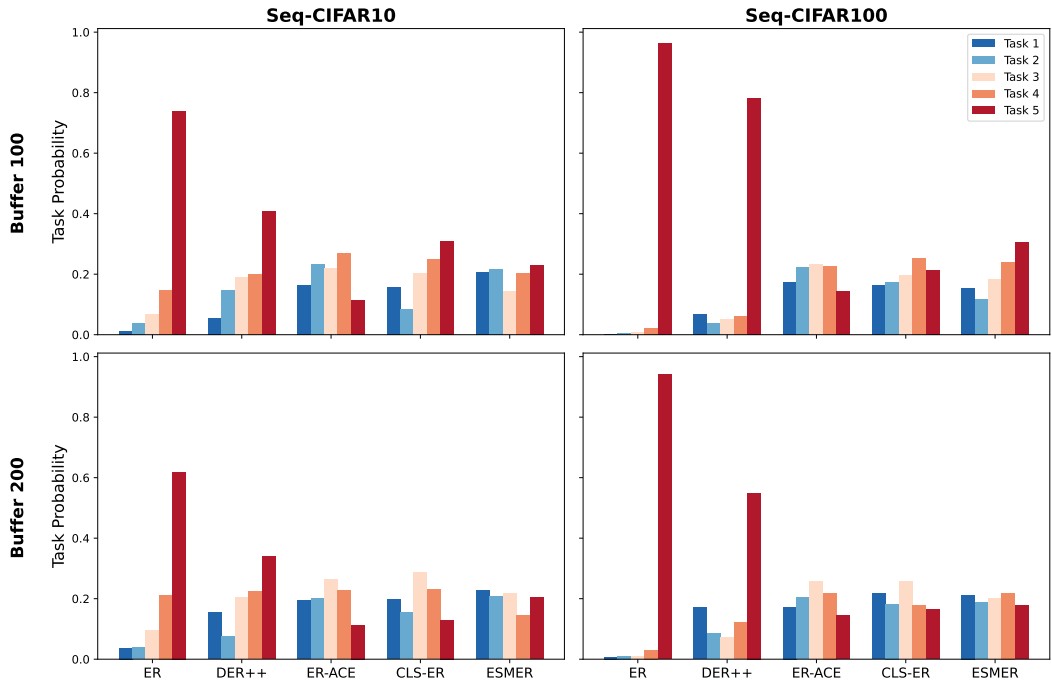

Figure S2: Average probability of predicting each task at the end of training for different datasets and buffer sizes. ESMER provides more uniform predictions.

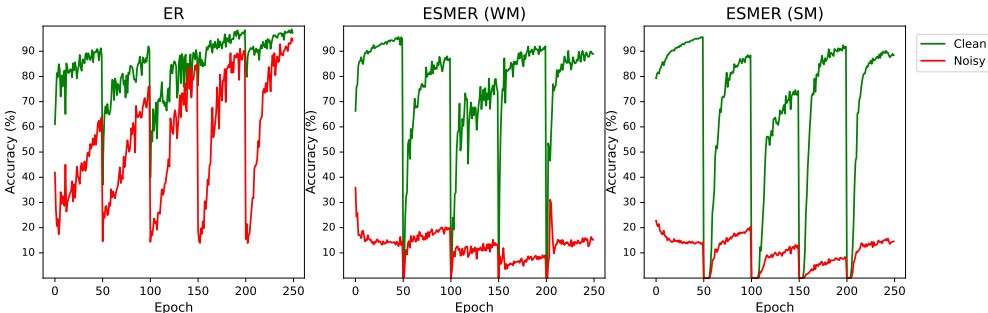

Figure S3: Accuracy of the models on clean and noisy samples as training progresses. We compare ER with the working model (WM) and stable model (SM) in ESMER on Seq-Cifar10 with 50% symmetric noise and 200 buffer size. The working model initially has slightly higher memorization than SM but then recovers with the aid of the consistency loss from the stable model.

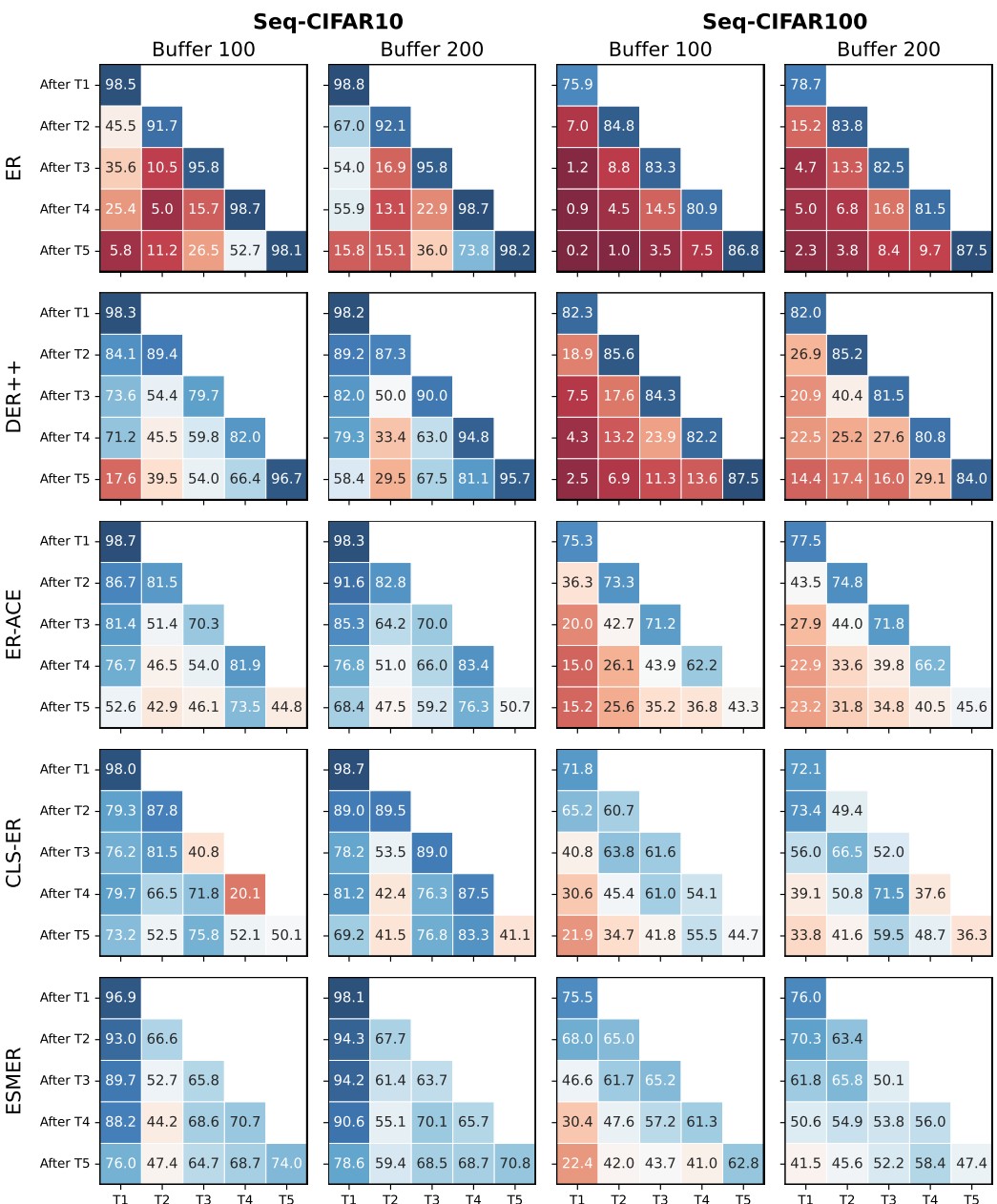

Figure S4: Taskwise performance of the models (x-axis) after training on each task on different datasets with varying memory budget. ESMER provide a good tradeoff between the stability and plasticity of the model.

