# OpenReview forum: "Error Sensitivity Modulation based Experience Replay: Mitigating Abrupt Representation Drift in Continual Learning"
_ICLR.cc/2023/Conference — ICLR 2023 poster_

### Official Review · Reviewer_pRxq · 2022-10-20

**Confidence:** 3
**Correctness:** 3
**Technical Novelty And Significance:** 3
**Empirical Novelty And Significance:** 3
**Recommendation:** 6

**Clarity, Quality, Novelty And Reproducibility:**

The paper is reasonably understandable, but a few points should be clarified:

- Generally: It took me a while to understand that the paper uses the word "memory" for what seems to be networks, or learners. This is confusing because this term is usually associated with buffers of items (samples or  returns).

In fact, the paper itself actually seems to use "semantic memory" in both senses !  E.g. in the line just after Eq. 3, where it seems to apply to a buffer of l_s values.

I would strongly recommend using the term "network" or "learner" to refer to the actual networks theta_s and theta_w,  and reserve the word "memory" for buffers of samples or losses.


- p. 5: "a task warm-up period during which the running estimate is not updated"

Does this imply that even though the method doesn't require access to task ID, it does require access to task boundaries / notifications of task changes? If so, this should be explicitly mentioned in the introduction.


- Greek 'alpha' and lowercase Latin 'a' in eq. 6 are very similar and a bit confusing! Maybe some different letters could be used?

- Please do explain acronyms! For example, in Class-IL / Task-IL - what is "IL" ? (I know it's "incremental learning", but it should really be explicitly stated in the paper, unless I missed it). Also, in the Related Work, what do ER and DER stand for ?

- The paper explains the various kinds of Class-IL, but not Task-IL, which is used in Table 1 (unless I missed it?)

- Figure 4a: Can you confirm whether the red curve shows "accuracy" with regard to the wrong/noised label, and therefore lower red curves are better? If soe this should be explicitly stated somewhere.


**Strength And Weaknesses:**

- Strengths:

The method seems novel and interesting, and the results are promising.

- Weaknesses:

One possible concern is that all baselines used for comparison are variants of the ER method. Since this is a bit beyond my  field I can't really assess whether this is an adequate representation of existing replay-based methods. If this is acceptable to other reviewers, then I can't see anything much wrong with the substance of paper.

As a minor point, the presentation is at times a bit confusing and could be improved, see below.

**Summary Of The Paper:**

= Update after author response =

I thank the authors for their clarifications. I maintain  my recommendation for acceptance.

= Original review =

The paper proposes a new approach to replay-based continual learning. The purpose of this approach is to prevent catastrophic representation drift caused by the large errors that occur when new tasks are introduced.

The approach maintains a slow-changing version  of the main learning network, clips sample losses if they are too far above a running average of previous losses (as estimated by the slow-changing network), penalizes disagreement with the slow-changing version for replay loss, and actually uses the slow-changing network at inference.

The method is compared to different variants of the so-called ER method  (another replay-based method) and found to improve performance.

**Summary Of The Review:**

This new method for improving stability of continual learning seems interesting and promising. If more knowledgeable reviewers are satisfied that the set of baseline methods used for performance comparison is representative, I have no objection to acceptance (pending clarifications as requested above).

---

> ### Author Response · Authors · 2022-11-11
> **Response to Reviewer pRxq**
>
> *Thank you for your valuable suggestions and feedback. We are glad you found the method interesting. Please see the common response and revised manuscript for the additional results.*
>
> > One possible concern is that all baselines used for comparison are variants of the ER method. Since this is a bit beyond my field I can't really assess whether this is an adequate representation of existing replay-based methods. If this is acceptable to other reviewers, then I can't see anything much wrong with the substance of paper.
>
> We based our comparison on state-of-the-art replay-based approaches, as they currently present an effective approach for mitigating catastrophic forgetting in challenging CL settings where non-replay-based methods generally fail. Earlier works have shown the superior performance of the replay-based method over non-replay-based methods [P1,P2]. Furthermore, the main goal of our work is to demonstrate the effectiveness of error sensitivity modulation in mitigating the abrupt drift in representations that ER methods undergo. Therefore, we evaluate our method under challenging lower buffer regimes.
>
> Among the replay-based methods, we consider both single and multiple memory systems. We compare with CLS-ER which employs two semantic memories, and the performance improvement of ESMER over CLS-ER shows that error sensitivity modulation allows for efficient utilization of memory. This is further supported by our ablation study (Table 3), whereby adding ESM and error-sensitive sampling (row 1) considerably improves the performance of the dual memory system alone (row 3).
>
> Please let us know if we can provide more information. We would be happy to discuss this further.
>
> > As a minor point, the presentation is at times a bit confusing and could be improved, see below.
>
> We apologize for the confusion and appreciate your valuable suggestions. We have revised the manuscript with the following changes:
>
> - For clarity, we renamed the **semantic memory** to the **stable model**.
> - The task warmup stage does make the knowledge about the task boundary necessary, we clarified this in the introduction and made it more clear in the methodology. *While we use the task boundary for training, we do not require task labels at inference*.
> - We changed the letter for denoting the sampled number in Eq. 6 from $a$ to $u$.
> - We added the full forms of the acronyms and the explanation for Task-IL in the caption of Table 1 in the main paper and more details in the Experimental Details section A.1 in Appendix.
> - We added more explanation in the caption of Figure 4.
>
> *Please let us know if these changes make it clearer. We would be happy to make further changes to make the manuscript clearer.*

---

> > ### Comment · Reviewer_pRxq · 2022-11-30
> > **Response**
> >
> > As mentioned in my updated review above, I thank the authors for their clarifications and maintain my recommendation for acceptance.

---

> > > ### Author Response · Authors · 2022-12-08
> > > **Thank you**
> > >
> > > Thank you for your valuable feedback and suggestions to improve the paper. We really appreciate your support in accepting the paper.

---

> ### Author Response · Authors · 2022-11-29
> **Response to Reviewer's Update**
>
> > = Update after author response =
> >
> > I thank the authors for their clarifications. I maintain my recommendation for acceptance.
>
> Thank you, we are glad we were able to clarify. Could you please let us know if there remain any concerns we could address or provide more details that would help increase your confidence in the paper?

---

### Official Review · Reviewer_ZgEC · 2022-10-23

**Confidence:** 4
**Correctness:** 4
**Technical Novelty And Significance:** 3
**Empirical Novelty And Significance:** 3
**Recommendation:** 8

**Clarity, Quality, Novelty And Reproducibility:**

Clarity: The paper is well written and it’s very clear.

Quality: The paper is of high quality with a well-motivated method and good support by experiments.

Novelty: The method is novel. I believe the concept of error sensitivity will be inspiring to the continual learning community.

Reproducibility: The authors promise to release the code and most details are well documented in the paper and the appendix.


**Strength And Weaknesses:**

Strength:
1. The proposed method considers the large errors incurred by data from new tasks as a potential reason for abrupt presentation drift in continual learning. Based on this insight, the paper proposes to downweigh the data points with large errors to reduce the drift and the experiments show that it is very effective.
2. The experiments show that the performance is strong, surpassing CLS-ER even with one fewer model to keep track of. (Only the stable model instead of both plastic and stable models in CLS-ER)
3. The experiments on continual learning with noisy labels are interesting.
4. The paper is well-written and easy to follow.

Weakness:
1. The proposed method is only tested on small and relatively simple datasets like CIFAR10 and CIFAR100. I think the results are more complete and convincing if results on tiny/Mini/Full ImageNet are also reported and compared.
2. The proposed method has high complexity and many hyperparameters.


**Summary Of The Paper:**

This paper presents a new experience replay method using error sensitivity as a modulation method in continual learning. The proposed method keeps track of the classification loss during continual learning and when the newly-received data incurs a high classification loss, it gets downweighed to reduce its effect. In this way, the learned representations are more stable and suffer less from abrupt drift during task transition. The paper also adopts a semantic memory (a momentum-updated model) to further stabilize the continual learning process, following a prior work, CLS-ER. Experiments show that the proposed method achieves better performance in both standard continual learning benchmarks and settings with label noise.

**Summary Of The Review:**

This is a great paper presenting a novel method and strong results. It would be better if more experiments can be performed on some more complex datasets.

---

> ### Author Response · Authors · 2022-11-11
> **Response to Reviewer ZgEC (2/2)**
>
> > The proposed method has high complexity and many hyperparameters.
>
> **Complexity:**
>
> While the additional semantic memory in ESMER introduces a memory overhead, the main goal of our work was to demonstrate the effectiveness of error sensitivity modulation (ESM) in mitigating the abrupt drift in representations. We further show in our ablation study that the dual memory setup presents a promising approach for consolidating knowledge in CL. We compared our method with CLS-ER, which employs two semantic memories (the stable model and the plastic model), and the performance improvement of ESMER over CLS-ER shows that error sensitivity modulation allows for efficient utilization of memory. This is further supported by our ablation study (Table 3), whereby adding ESM and error-sensitive sampling (row 1) considerably improves the performance of the dual memory system alone (row 3).
>
> Compared to CLS-ER, ESMER has much lower memory and computational overhead. Please note that for inference we only use semantic memory hence our method does not incur any additional cost at inference compared to other methods.
>
> We also conducted another ablation study whereby we only Error sensitivity modulation on top of the baseline ER method and show considerable improvement in performance. For 25% label noise, the results improve **from 25.54 to 38.50**. The empirical results show the effectiveness of error sensitivity modulation even in a single memory system and provide further credence to our findings.
>
> Future work in making semantic memory smaller and different mechanisms for encoding semantic knowledge is an interesting avenue of research.
>
> **Hyperparameters:**
>
> While our method does introduce hyperparameters, we would like to emphasize that they are complementary in nature and the majority of them can be fixed. In Section A.1.1, we provide details of our hyperparameter tuning approach. Furthermore, Section A.5 shows that the hyperparameters are complementary in nature, which allows us to fix the majority of parameters. This significantly facilitates hyperparameter tuning. Also, our method is not very sensitive to a particular choice of parameters (Table S4).
>
> For our experiments, we fixed the alpha parameters for the semantic memory and error memory to 0.999 and 0.99 and observed that the $\gamma$ parameter does not require much tuning and can be fixed. The parameter that has the most significant impact on results is the update frequency, $r$, followed by the loss margin parameter $\beta$. Table S.1 shows our method doesn't require much tuning for different buffer sizes.
>
> Compared to CLS-ER, ESMER is much easier to tune to different datasets as only one semantic memory is needed.
>
> *Please let us know if we failed to address your concerns with the new experiments and explanations. We would be happy to engage in a discussion and provide more details.*
>
> **Reference:**
>
> [P1]  Arani, et al. "Learning Fast, Learning Slow: A General Continual Learning Method based on Complementary Learning System." ICLR 2021.

---

> > ### Comment · Reviewer_ZgEC · 2022-12-01
> > **Thanks for the response**
> >
> > Thanks for the response, and they have addressed my concerns. I will keep my rating.
> >
> > I encourage authors to complete their experiments on TinyImageNet further. Also, I know most of the model complexity is incurred by Mean-ER (CLS-ER), and thus it's good to see results based on ER baseline.
> >
> > Also, I am happy to see the term "semantic memory" has been changed according to other reviewers' comments. It was also confusing for me.

---

> > > ### Author Response · Authors · 2022-12-08
> > > **Thank you**
> > >
> > > We are really glad we were able to address your concerns and that you approve of the changes. Thank you for the valuable suggestions and feedback for improving our paper further, we will try to incorporate all of them.

---

> ### Author Response · Authors · 2022-11-11
> **Response to Reviewer ZgEC (1/2)**
>
> *Thank you for your encouraging remarks and valuable feedback. We are delighted that you believe error sensitivity modulation will be inspirational to the research community of lifelong learning. This is the key takeaway message of our study. Please see the common response for an overview of the new experiments and sections in the revised manuscripts. We attempt to address each of your points individually below.*
>
> > The proposed method is only tested on small and relatively simple datasets like CIFAR10 and CIFAR100. I think the results are more complete and convincing if results on tiny/Mini/Full ImageNet are also reported and compared.
>
> Thank you for bringing this to our attention. We ran our method on Seq-TinyImageNet with a 500 buffer size. The results are quite promising given that CLS-ER achieves this result with two additional memories and their single semantic memory variant (Mean-ER in Table S2 in P1) achieves 24.97%. Please note that because of the time constraint, we couldn’t perform hyperparameter tuning, and it is possible to find better parameters for this setting. We wanted to engage in the discussion earlier instead of waiting for results. Baseline results from [P1].
>
> | Method   | Seq-TinyImageNet |
> |-------------:|:------------:|
> | ER          | 9.99 ± 0.29 |
> | DER++   | 19.38 ± 0.56 |
> | CLS-ER  | 31.03 ± 0.56 |
> | ESMER  | **31.69 ± 0.42** |
>
> We would also like to emphasize that while the CIFAR10 is a simple dataset, CIFAR100 actually presents a challenging recognition task with a high semantic similarity between classes and a low number of samples per class. The smaller image size and dataset size make the experimentation substantially easier. On top of the underlying dataset, we simulate several challenging CL scenarios. To further simulate the effect of longer task sequences,  we ran Cifar100 for 10 tasks (10 classes each) and 20 Tasks (5 classes). As the number of tasks increases, the CL task becomes more challenging as the method has to consolidate across a higher number of previous tasks.  ESMER consistently provides generalization gains.
>
> | Method    |   5 Tasks   | 10 Tasks | 20 Tasks |
> |--------------|:--------------:|:------------:|:------------:|
> | ER           | 21.91±0.36 | 14.17±0.53 | 9.97±0.68   |
> | DER++    | 30.68±1.35 | 25.50±3.07 | 20.50±1.42 |
> | ER-ACE  | 35.17±1.17 | 25.75±1.56 | 18.68±0.82 |
> | CLS-ER  | 43.80±1.89 | 35.42±0.47 | 25.98±1.70 |
> | ESMER  | **48.77±0.31** | **36.37±0.46** | **27.26±0.50** |
>
> We also evaluated the effect of label noise on CIFAR100. ESMER provides considerable gains over baselines under this challenging setting.
>
> | Method    |   Clean      | 10%        | 25%         | 50%       |
> |--------------|:--------------:|:------------:|:------------:|:------------:|
> | ER           | 21.91±0.36 | 17.12±0.36 | 13.00±0.42   | 8.01±0.07 |
> | DER++    | 30.68±1.35 | 25.13±1.07 | 18.99±0.62 | 11.04±0.42 |
> | ER-ACE  | 35.17±1.17 | 23.77±1.14 | 13.59±0.66 | 5.71±0.33 |
> | CLS-ER  | 43.80±1.89 | 35.19±0.85 | 24.59±0.35 | 12.16±0.69 |
> | ESMER  | **48.77±0.31** | **41.80±0.88** | **33.49±1.29** | **20.68±1.21**|

---

### Official Review · Reviewer_CEB8 · 2022-10-25

**Confidence:** 5
**Correctness:** 3
**Technical Novelty And Significance:** 3
**Empirical Novelty And Significance:** 3
**Recommendation:** 5

**Clarity, Quality, Novelty And Reproducibility:**

Clarity needs to be improved. For example, the Fig 1 have notations that are not introduced before the figure so it’s hard to parse the information from it.  It is not clear how the hyperparameters in this approach can be optimized for the continual learning setting when considering different datasets.

**Strength And Weaknesses:**

Strengths:
The idea of modulating the learning based on the error is well motivated and using it to create an episodic memory buffer seems improve the continual learning accuracy in the clean as well as label corruption settings

Weaknesses:

•	There is a significant overhead introduced by the dual memory mechanism so it will be fair to compare with other single memory approaches by maintaining a fixed overall memory.

•	In general, the related works need to be more comprehensive and explicit in explaining how the proposed work is different from the literature and compare if needed [6] : The idea of error sensitivity-based modulation seems to be explored before in the local learning-based approaches [1,2,3]. Why only compare with replay-based approached when comparing the performance. Other non-replay-based approaches have shown superior performance and have the advantage of not requiring the memory buffer [2,4].

•	Not clear if the improved results hold when a larger memory buffer (eg., 5k) is used.

•	Only a single metric (accuracy) is used for comparison. Other metrics such as average forgetting [5] shall be used to

•	Effect of task sequence not considered! do the results hold when the task sequence is changed?

•	Why does ESMER have lower accuracy (on test data) for the task it is trained on? As seen in figure 3 diagonal elements.

•	The ablation does not show the effect of just keeping the error sensitivity modulation without the semantic memory or reservoir sampling.

•	Do the results on label corruption (fig2) hold for Cifar-100 data as well?

•	Does this approach work in the online continual learning setting?

[1] Dellaferrera, G., & Kreiman, G. (2022). Error-driven Input Modulation: Solving the Credit Assignment Problem without a Backward Pass. arXiv preprint arXiv:2201.11665.

[2] Madireddy, S., Yanguas-Gil, A., & Balaprakash, P. (2020). Neuromodulated neural architectures with local error signals for memory-constrained online continual learning. arXiv preprint arXiv:2007.08159.

[3] Kudithipudi, Dhireesha, et al. "Biological underpinnings for lifelong learning machines." Nature Machine Intelligence 4.3 (2022): 196-210.

[4] Li, S., Du, Y., van de Ven, G. M., & Mordatch, I. (2020). Energy-based models for continual learning. arXiv preprint arXiv:2011.12216.

[5] Mai, Zheda, et al. "Online continual learning in image classification: An empirical survey." Neurocomputing 469 (2022): 28-51.

[6] Pham, Q., Liu, C., & Hoi, S. (2021). Dualnet: Continual learning, fast and slow. Advances in Neural Information Processing Systems, 34, 16131-16144.

**Summary Of The Paper:**

This work builds upon the notion of a complementary learning system that consists of multiple memories: episodic memory and the semantic memory in this case. Although this notion is utilized in many works, the main contribution of this work is in designing the episodic memory to contain samples from the current batch that are pre-selected based on their distance from the mean of an errors of all samples along the training trajectory. These errors are also used to modulate the learning. The results on Seq-CIFAR-10, Seq-CIFAR-100 and GCIL show good accuracies in low memory buffer scenarios as well as in the presence of label corruptions.

**Summary Of The Review:**

•	The related works and consequently the approaches compared needs to be updated.

•	The effect of increase in the memory for dual memory approaches need to be considered when comparing with other approaches.

•	Alternative continual learning metrics and task sequence need to be considered.

---

> ### Author Response · Authors · 2022-11-11
> **Response to Reviewer CEB8 (4/4)**
>
> > There is a significant overhead introduced by the dual memory mechanism so it will be fair to compare it with other single-memory approaches by maintaining a fixed overall memory.
>
> While it is true that semantic memory introduces a memory overhead, the main goal of our work is to demonstrate the effectiveness of error sensitivity modulation (ESM) in mitigating the abrupt drift in representations. We compare with CLS-ER which employs two semantic memories (the stable model and the plastic model), and the performance improvement of ESMER over CLS-ER shows that error sensitivity modulation allows for efficient utilization of memory. This is further supported by our ablation study (Table 3), whereby adding ESM and error-sensitive sampling (row 1) considerably improves the performance of the dual memory system alone (row 3).
> We would like to emphasize that another key aspect of a CL method is also how much it can retain knowledge with limited data available, as storing data from the previous task is not only a matter of available memory but can also cause privacy issues. By mitigating the representation drift at the task boundary and aggregating knowledge in the semantic memory, ESMER can retain much more information. As a reference, Table 1 shows that even with twice the number of samples (200 buffer size), single memory approaches (ER, DER++, and ER-ACE) are unable to surpass the performance of ESMER with half the number of samples (100 buffer size).
>
> > In general, the related works need to be more comprehensive and explicit in explaining how the proposed work is different from the literature and compare if needed [6]: The idea of error sensitivity-based modulation seems to be explored before in the local learning-based approaches [1,2,3].
>
> Thank you for sharing these papers. Please note that we highlight a key characteristic of the brain: that it learns more from small errors compared to large errors, unlike standard ANNs where learning scales linearly with the error size. Subsequently, the error sensitivity modulation in our paper specifically focuses on utilizing a memory of errors to modulate the contribution of samples to learning based on their error magnitude, such that the model learns more from small errors.
>
> Please correct us if we missed something. To the best of our knowledge [1–3], we do not employ any such  mechanism to bias learning toward small errors. [1] replaces the backward pass by perturbing the input signal based on error-related information and utilizing the difference between the network response to the input and to its perturbed version to compute the synaptic updates. As such, it provides an alternative to backward propagation and doesn’t modify the degree of learning from each sample in the batch based on how far the error is from the mean statistics of previous errors. Neuromodulation [2,3] involves modifying the rate of plasticity of the individual model weights. Therefore, these methods can be considered closer to regularization-based methods where weight changes on a subset of the network are penalized, which are considered important for previous tasks. We have added Section A.2 in the Appendix, highlighting the difference between error sensitivity modulation in our approach and neuromodulation and metaplasticity.
>
> *We hope to have addressed your main concerns with the new experiments and explanations. We would be happy to engage in a discussion, address your concerns, and provide more details.*
>
> **Reference:**
>
> [P1] Buzzega, et al. "Dark experience for general continual learning: a strong, simple baseline." NeuRIPS 2020
>
> [P2] van de Ven, et al. "Three scenarios for continual learning." arXiv e-prints 2019
>
> [P3] Caccia, et al. "New Insights on Reducing Abrupt Representation Change in Online Continual Learning." ICLR 2021
>
> [P4] Arani, et al. "Learning Fast, Learning Slow: A General Continual Learning Method based on Complementary Learning System." ICLR 2021
>
> [P5] Sarfraz, et al. "SYNERgy between SYNaptic consolidation and Experience Replay for general continual learning." CoLLAs 2022

---

> ### Author Response · Authors · 2022-11-11
> **Response to Reviewer CEB8 (3/4)**
>
> > Only a single metric (accuracy) is used for comparison. Other metrics such as average forgetting [5] shall be used to
>
> For empirical evaluation, we use the average performance across the tasks, which is the most common metric in the CL literature and takes into account both the stability and plasticity of the model. Additionally, we conduct behavioral analysis of the different models and compare representation drift, task recency bias, CLS-ER task-wise performance during training, and memorization of noisy labels.
> Forgetting is another pertinent metric. We didn’t include forgetting because the common implementation assumes that the model has the highest accuracy on a task right after it is trained on it. However, this is not always the case with the semantic memory (which is used for inference) in ESMER. As the semantic memory aggregates the knowledge in the working model (which has the highest performance at the end of task training, see Figure S1) using an exponential moving average, the semantic memory achieves the highest accuracy on the previous tasks while learning the new task as it slowly adapts to the weights of the working model. Thus, forgetting metric would be misleading for ESMER. For instance, Tasks 3 and 4 in Figure S1 Seq-CIFAR100 on buffer size 200 give the wrong impression that ESMER has a positive backward transfer. Therefore, ESMER would have much lower values for forgetting compared to single memory baselines, but it would be misleading.
>
> Ideally, if we had the accuracy of the semantic memory at each iteration during training and not just at the task boundaries, we could use the maximum performance of each task as a reference to calculate forgetting. However, this is computationally infeasible. This applies to both ESMER and  as they use the EMA models for inference.
>
> One possible alternative that doesn’t require retraining the models and monitoring the performance of the semantic memory at each epoch or iteration is to use the maximum performance of the tasks as the reference. For instance, given the task-wise performance matrices in Figure 3, for ESMER Task 3, we can use 70.1 as a reference instead of 63.7. Using this approach to evaluate forgetting for the methods in Figure 3 would provide the following forgetting metrics: ER (61.18), DER++ (33.45), CLS-ER (23.48), and **ESMER (7.33)**. While this bids well for our method, it might be misleading.
>
> If you have any suggestions or comments, we would be happy to discuss them.
>
> > Why does ESMER have lower accuracy (on test data) for the task it is trained on? As seen in figure 3 diagonal elements
>
> Sorry for the confusion on this. For inference, we use semantic memory. As mentioned earlier, it aggregates the knowledge in the working model using an exponential moving average, and the semantic memory achieves the highest accuracy on the previous tasks while learning the new task as it slowly adapts to the weights of the working model. Figure S1 provides side-by-side task-wise performance of the working model and the semantic memory.
>
> > Why only compare with replay-based approached when comparing the performance. Other non-replay-based approaches have shown superior performance and have the advantage of not requiring the memory buffer [2,4].
>
> We based our comparison on state-of-the-art replay-based approaches, as they currently present an effective approach for mitigating catastrophic forgetting in challenging CL settings where non-replay-based methods generally fail. Earlier works have shown the superior performance of the replay-based method over non-replay based methods [P1,P2].
>
> EBM [4] does not perform comparable to replay-based methods and their main comparison is with non-replay-based method. For instance, they report 38.84% accuracy on CIFAR-10,  while ESMER (with a 100 buffer size) achieves 65.37%. In Table 6 of their paper, the authors also provide results for the replay-based variant of their method with much higher buffer sizes. Even with 10x samples (1000 buffer size), they report 44.76% accuracy, which is much lower than what ESMER achieves with 1/10th samples (100 buffer size).
>
> While it is true that non-replay-based methods have the advantage of not requiring buffer samples, they struggle to compete in performance with replay-based methods. We would be happy to discuss this further.

---

> ### Author Response · Authors · 2022-11-11
> **Response to Reviewer CEB8 (2/4)**
>
> > Not clear if the improved results hold when a larger memory buffer (eg., 5k) is used.
>
> In our study, we focused on a low buffer regime as it presents a more challenging and realistic CL setting. It also tests how well a method retains knowledge with a very limited number of samples from previous tasks (e.g., a buffer size of 100 corresponds to ~1 sample for the previous 80 classes while learning the last task in Seq-CIFAR100).
>
> Moreover, [P3] showed that ER methods undergo an abrupt drift in representations at the task boundary, and while methods can recover from initial disruptions with larger buffer sizes, under lower buffer regimes they fail to do so. Since our method aims to mitigate the representation drift, we focus our empirical evaluation on challenging lower buffer regimes, even extremely small sizes. The empirical findings indicate that error sensitivity modulation effectively mitigates disruptions and reduces task interference to retain more knowledge from previous tasks.
>
> For completion, we ran our method on S-CIFAR10 with a 1000 buffer size, and it achieves 76.25±0.91,  which is comparable to the baselines [p5]. Please note that because of the time constraint, we couldn’t perform hyperparameter tuning, and it is possible to find better parameters for this setting. We wanted to engage in the discussion earlier instead of waiting for results. Note that for such large buffer sizes, all methods start converging to similar performance as they have arguably enough samples of previous classes to learn the joint distribution and may not require measures to avoid forgetting. This makes it challenging to distinguish the effectiveness of CL methods.
>
> > Does this approach work in the online continual learning setting?
>
> This is a good suggestion. While it was not the goal of our study and is beyond the scope of the paper, we believe that our method can be extended to the online continual learning setting, which we leave for future work.
>
> > Clarity needs to be improved. For example, Fig 1 has notations that are not introduced before the figure so it’s hard to parse the information from it.
>
> We wanted Figure 1 to act as a reference throughout the paper. The description of the figure and the process provides a general overview for the first pass, and once the reader goes through the methodology, having the loss term notations there makes it more intuitive and easier to understand. If you consider that this adds more to the confusion than it helps with understanding, we can remove the loss terms.
>
> > It is not clear how the hyperparameters in this approach can be optimized for the continual learning setting when considering different datasets.
>
> In Section A.1.1, we provide details of our hyperparameter tuning approach. Furthermore, Section A.5 shows that the hyperparameters are complementary in nature, which allows us to fix the majority of parameters. This significantly facilitates hyperparameter tuning. Also, our method is not very sensitive to a particular choice of parameters. Please let us know if these two sections provide the required information. We would be happy to provide more details.

---

> ### Author Response · Authors · 2022-11-11
> **Response to Reviewer CEB8 (1/4)**
>
> *Thank you for the valuable suggestions. We are glad that you consider the main idea of our approach well-motivated. We attempt to address each of your points individually.*
>
> > Effect of task sequence not considered! do the results hold when the task sequence is changed?
>
> Thank you for the valuable suggestion. To evaluate the effect of task sequence, we vary the number of tasks in Seq-CIFAR100. In addition to the 5 tasks with 20 classes in the main table, we also consider 10 and 20 tasks with 10 and 5 classes in each task, respectively. ESMER consistently provides generalization gains. Please let us know if this addresses your concern.
>
> | Method    |   5 Tasks   | 10 Tasks | 20 Tasks |
> |--------------|:--------------:|:------------:|:------------:|
> | ER           | 21.91±0.36 | 14.17±0.53 | 9.97±0.68   |
> | DER++    | 30.68±1.35 | 25.50±3.07 | 20.50±1.42 |
> | ER-ACE  | 35.17±1.17 | 25.75±1.56 | 18.68±0.82 |
> | CLS-ER  | 43.80±1.89 | 35.42±0.47 | 25.98±1.70 |
> | ESMER  | **48.77±0.31** | **36.37±0.46** | **27.26±0.50** |
>
> > The ablation does not show the effect of just keeping the error sensitivity modulation without the semantic memory or reservoir sampling.
>
> Thank you for pointing this out. We added this ablation study to Table 4. Adding error sensitivity modulation on top of ER improves the performance in all the considered scenarios. For 25% label noise, the results improve from 25.54 to 38.50. The empirical results show the effectiveness of error sensitivity modulation even in a single memory system and provide further credence to our findings.
>
> | ESM | SM    | ESRS | B = 50 | B = 100 | B = 200 | N = 25% | N = 50% |
> |--------|:-------:|:--------:|:-----------:|:------------:|:------------:|:-----------:|:------------:|
> | ✓     | ✓      | ✓        | **59.94±1.67** | **65.37±0.68** | **69.16±0.54** | **56.61±1.40** | **42.43±2.89** |
> | ✓     | ✓      | ✗        | 57.53±1.80 | 64.44±0.62 | 69.07±0.87 | 51.16±1.56 | 34.42±2.79 |
> | ✗     | ✓      | ✗        | 37.11±4.88 | 51.59±5.26 | 63.41±1.43 | 38.84±0.52 | 22.10±2.62 |
> | ✓     | ✗      | ✗        | 36.03±0.83 | 42.56±0.56 | 54.58±2.89 | 38.50±1.53 | 24.70±1.49 |
> | ✗     | ✗      | ✗        | 32.51±1.77 | 41.10±0.10 | 44.79±1.86 | 25.54±2.14 | 19.72±1.53 |
>
> ESM: error sensitivity modulation, SM: stable model, ESRS: error sensitive reservoir sampling, B: buffer size, N: label noise
>
> > Do the results on label corruption (fig2) hold for Cifar-100 data as well?
>
> Thank you for the valuable suggestion. We conducted the label corruption analysis on Seq-CIFAR100 and the empirical results show the effectiveness of ESMER in this challenging setting.
>
> | Method    |   Clean      | 10%        | 25%         | 50%       |
> |--------------|:--------------:|:------------:|:------------:|:------------:|
> | ER           | 21.91±0.36 | 17.12±0.36 | 13.00±0.42   | 8.01±0.07 |
> | DER++    | 30.68±1.35 | 25.13±1.07 | 18.99±0.62 | 11.04±0.42 |
> | ER-ACE  | 35.17±1.17 | 23.77±1.14 | 13.59±0.66 | 5.71±0.33 |
> | CLS-ER  | 43.80±1.89 | 35.19±0.85 | 24.59±0.35 | 12.16±0.69 |
> | ESMER  | **48.77±0.31** | **41.80±0.88** | **33.49±1.29** | **20.68±1.21**|

---

> ### Author Response · Authors · 2022-11-30
> **A gentle reminder**
>
> Hi, could you please let us know if our response, additional experiments, and revisions in the manuscripts addressed your concerns? Based on your feedback and suggestions, we performed experiments on S-CIFAR-100 with different sequence lengths and noisy labels and ran S-CIFAR-10 with a higher buffer size, and performed ablation with Error Sensitivity Modulation alone on top of ER. The empirical results of the new experiments further support our claims and we hope that these will address your major concerns. We would be happy to address your remaining concerns and hope to increase your confidence in the acceptance of our paper.

---

### Author Response · Authors · 2022-11-11
**General Response**

We thank the reviewers for their valuable feedback and suggestions. We tried to incorporate the suggestions into the revised manuscript and ran additional experiments. Here we provide a summary of the revisions:

**New Experiments:**

- **Effect of longer task sequences on Seq-CIFAR100 (5, 10, and 20 Tasks) in Table 2:** Results demonstrate the effectiveness of ESMER in consolidating knowledge over longer task sequences, as well.

- **Effect of adding degrees of label noise in Seq-CIFAR100 (10%, 25%, and 50%) in Table 3:** Esmer is considerably more robust to label noise compared to baselines.

- **Extended ablation study with the effect of adding error sensitivity modulation on top of baseline ER:** The empirical results show the effectiveness of error sensitivity modulation even in a single memory system. For 25% label noise, the results improve *from 25.54 to 38.50*.

**New Section and Modifications:**

-  We added a section comparing the error sensitivity modulation in our paper with Neuromodulation and Metaplasticity in Section A.2

- Based on the suggestions of the reviewers, we made minor changes to improve the clarity of the paper.

All the modifications are in blue color in the paper.

---

### Author Response · Authors · 2022-11-17
**Gentle reminder for end of discussion period**

As the end of discussion period approaches, we would like to have the opportunity to engage with all the reviewers and address any remaining concerns. We tried to address each reviewer's raised concerns, conducted many additional experiments, and revised the manuscript based on their suggestions. We would be happy to know if we were able to address your concerns and discuss them further to provide further information and clarifications if needed.

---

### Decision · Program_Chairs · 2023-01-20

**Decision:**

Accept: poster

**Justification For Why Not Higher Score:**

This is a good paper. I am recommending a poster since the results are only based on cifar-10 and cifar-100. It is a promising approach that needs more work to become impactful.

**Justification For Why Not Lower Score:**

There is no reason to reject this paper. It is a solid paper that is worth publishing.

**Metareview: Summary, Strengths And Weaknesses:**

This paper proposes a new replay-based method for continual learning. The proposed ESMER is a dual memory rehearsal-based system which has a working model, a stable model, and a buffer which incorporates error-sensitivity-based reservoir sampling. The proposed algorithm works better than the existing replay-based methods for CL.

Reviewers asked several questions in their reviews and the authors provided convincing answers to all of them. The authors also added more experimental results that sound convincing. I recommend an acceptance for this paper.

However, I strongly encourage the authors to add imagenet experiments to the final version of the paper since cifar-10 and cifar-100 are totally not sufficient for current standards in continual learning.

**Note From Pc:**

if the above contains the word "oral" or "spotlight" please see: "oral" presentation means -> notable-top-5% and "spotlight" means -> notable-top-25%. As stated in our emails, we are disassociating presentation type from AC recommendations